# PD-L1 Is Involved in the Development of Non-Hodgkin’s Lymphoma by Mediating Circulating Lymphocyte Apoptosis

**DOI:** 10.3390/vaccines11091474

**Published:** 2023-09-11

**Authors:** Manal Mohamed Saber

**Affiliations:** Department of Clinical Pathology, Faculty of Medicine, Minia University, Minia 61519, Egypt; manal.saber@mu.edu.eg

**Keywords:** non-Hodgkin lymphoma, programmed death-ligand 1, apoptosis, circulating lymphocytes

## Abstract

Lymphocyte apoptosis plays a crucial role in tumor-induced immunosuppression. Programmed death ligand-1 (PD-L1) blocks lymphocyte activation via its receptor, PD-1. However, PD-L1/PD-1 expression and its role in enhancing immune suppression in non-Hodgkin lymphoma (NHL) have not been identified. The purpose of the study was to assess PD-L1/PD-1 expression in circulating lymphocytes in NHL and its role in immunosuppression. Twenty newly diagnosed NHL patients and twenty normal volunteers were enrolled in the study. PD-L1/PD-1 expression in circulating lymphocytes and the apoptosis of lymphocyte subsets were assessed using flow cytometry. The findings revealed that the PD-L1 expression in circulating CD3^+^, CD3^+^CD4^+^, CD3^+^CD8^+^, and CD20^+^ lymphocytes were dramatically upregulated in NHL patients (*p* < 0.001), whereas peripheral lymphocytes expressed low levels of PD-1. Compared with normal volunteers, a significant increase in lymphocyte apoptosis was revealed by annexin-V binding on T and B lymphocytes (*p* < 0.001). Peripheral lymphocytes expressing PD-L1 were four times more vulnerable to apoptosis than those expressing PD-1. Our findings imply that PD-L1 upregulation contributes to NHL development by promoting circulating lymphocyte apoptosis. This research adds to our understanding of the function of the PD-L1/PD-1 pathway in tumor evasion, establishing a novel therapeutic target in NHL. The results offer additional evidence for the immunomodulatory role of PD-L1 in circulating lymphocytes, providing a rationale for further investigations into immunological dysfunctions resulting from NHL. PD-L1^+^ lymphocytes could be employed as a biomarker to assess the effectiveness of immune systems and predict illness in patients with NHL.

## 1. Introduction

Non-Hodgkin’s lymphoma (NHL) tumors are the most prevalent hematological tumors in the world and constitute a heterogeneous entity [1]. The WHO has classified over eighty distinct NHL subtypes based on the cell type (NK, T, and B cells) and the clinical characteristics, genetic criteria, immunophenotyping, and morphology [2]. Significant insights into the interactions between lymphomas and the immune system have been attained [3]. Certain infections and immune deficiencies may play a role in NHL pathogenesis [4]. On the other hand, Lymphomas cause immunological alterations by lowering B lymphocytes and interrupting the regular interaction of B lymphocytes with other cells [5]. Immune-system silencing initiates the development and growth of NHL [6]. Immunosuppressive networks are characterized by cellular microenvironmental alterations and different signaling pathways [7]. The expression of programmed death-ligand 1 (PD-L1)/PD-1 could enhance tumor development via the inhibition of T-cell functions, indicating an immunosuppressive role of PD-L1/PD-1 [8,9,10].

PD-1 is an immunoglobulin superfamily member identified in B and T lymphocytes [11]. It significantly regulates peripheral tolerance [12]. Immune cells, tumor cells, and certain non-immune cells have exhibited PD-L1 expression [13,14]. Previous reports revealed PD-L1 and PD-1 expression in different types of NHL, including follicular lymphoma (FL), small B-cell lymphoma (SCL), T-cell lymphoma, and diffuse large B-cell lymphoma (DLBCL) [15]. PD-L1/PD-1 expression has been linked to tumor growth and a poor prognosis in NHL [16]. Therapeutic anti-PD-L1/PD-1 antibodies could reactivate the immune response against tumors; this may be a promising treatment for blood cancers such as lymphomas [17].

CD4^+^, CD8^+^, and CD20^+^ cells are types of immune cells that can eradicate cancer cells. When the immune system cannot function optimally, B and T cells fail to detect or destroy lymphoma cells [18,19]. Tumors develop different mechanisms, including inducing immune checkpoint dysregulation to bypass the immune system [20].

Given the importance of PD-L1/PD-1 signaling in immune regulation and the significance of immune responses in NHL, the PD-1/PD-L1 pathway might have a significant role in NHL development. This study assessed PD-L1/PD-1 expression as well as the functional features of circulating lymphocytes in NHL. The results demonstrate how immunosuppression is part of NHL and clarify PD-L1/PD-1 functions in circulating lymphocytes to predict and diagnose the disease.

## 2. Methods

### 2.1. Subjects

The study was conducted at the Clinical Pathology Department of the Faculty of Medicine, Minia University, Egypt. Recruitment of the study participants was performed via referral from oncology clinics and self-referral via leaflets distributed at other clinics. The subjects involved twenty newly diagnosed NHL patients and twenty normal controls. The healthy controls comprised ten males and ten females, with ages ranging from 25.5 to 50.0 years old. There were ten female and ten male NHL patients, of which seventeen were DLBCL patients, two were FL patients, and one was an SCL patient. NHL patients ranged in age from 27.5 to 49.5 years. NHL patients who lacked sufficient pathological or clinical data were excluded from the study. Subjects with autoimmune diseases, immune-system disorders, chronic infections, or who were using immunosuppressive drugs were excluded. Every subject signed a documented written consent form. This study followed the Helsinki Declaration and Good Practice Guidelines. The Ethics Committee of the Faculty of Medicine’s Institutional Review Board approved the study and the signed informed consent provided by NHL patients and normal controls (No. 253-2022).

### 2.2. Clinical Data

All subjects completed detailed history forms and underwent extensive clinical examinations that focused on monitoring lymphadenopathy and hepatosplenomegaly. Bone marrow aspiration and lymph node biopsies were conducted on each patient to determine the type, stage, assessment, and prognosis of their lymphoma. Three experienced lymphoma pathologists validated the pathological samples according to the classification of the WHO [21]. A flow cytometer was used for immunophenotyping. X-rays and urltrasound for abdomen and pelvis were performed on the patients. NHL was assessed using the Eastern Cooperative Oncology Group scale [22]. Subjects with insufficient pathological or clinical data were removed from the study; this included subjects with autoimmune diseases or chronic infections.

### 2.3. Antibodies

A CD4 monoclonal antibody (400 µg/mL, catalog no. 300506, BioLegend, San Diego, CA, USA); CD3 monoclonal antibody (200 µg/mL, catalog no. 300406, clone UCHT1, BioLegend); CD8 monoclonal antibody (100 µg/mL, catalog no. 344704, clone SK1, BioLegend); CD20 monoclonal antibody, (100 µg/mL, catalog no. 302304, clone 2H7, BioLegend); PD-L1 monoclonal antibody (400 µg/mL, catalog no. 309706, clone 29E.2A3, BioLegend); PD-1 monoclonal antibody (400 µg/mL, catalog no. 329906, clone EH12.2H7, BioLegend); and PE mouse IgG2a (k) isotype control (catalog no. 402203, BioLegend) were used in this study.

### 2.4. Flow Cytometry Analysis

Four tubes were identified for the PD-L1 expression: one for FITC-conjugated anti-CD3 and PE-conjugated anti-CD274; one for FITC-conjugated anti-CD4 and PE-conjugated anti-CD274; one for PE-conjugated anti-CD274 and FITC-conjugated anti-CD8; and one for FITC-conjugated anti-CD20 and PE-conjugated anti-CD274. Similar combinations of antibodies and PE-conjugated anti-CD279 were used for the PD-1 expression. PE isotype controls and unstained cells were used as negative controls. First, 100 μL of blood was placed into tubes, followed by 5 μL of monoclonal antibodies. The tubes were then vortexed and held for twenty minutes in the dark at an ambient temperature. Two milliliters of 1× lysing buffer (BioLegend, catalog no. 420301) were then added to the tubes. The contents of the tubes were mixed and maintained at room temperature for ten minutes. The tubes were then centrifuged for 5 min at 1200 rpm. Samples were washed using phosphate-buffered solution (PBS) (BioLegend, catalog no. 926201), and the supernatants were discarded. The cells were then resuspended in three hundred microliters of PBS. Finally, the cells were investigated using flow cytometry (BD Biosciences, San Diego, CA, USA).

At least 10,000 incidents were investigated. FACSDiva software was used for the data analysis. Forward-scatter versus side-scatter lymphocyte gating was used to exclude cell debris and aggregates. The results were displayed as proportions of stained cells with monoclonal antibodies. The percentage of positive cells was calculated based on the isotypic controls. Isotypic controls were used to calculate the cutoff values. A negative control (unstained cells) was used for each subject.

### 2.5. Apoptosis Measurements Using Flow Cytometry

Lymphocyte apoptosis was identified by flow cytometry analyses using an APC Annexin-V apoptosis detection kit (BioLegend, catalog no. 640930). First, the cells were stained for monoclonal antibodies. A cold staining buffer (BioLegend, catalog no. 420201) was used to wash the cells. The cells were then resuspended in an annexin-V binding buffer (BioLegend, catalog no. 422201). A total of 5 μL annexin-V (8 µg/mL) was added to each tube for fifteen minutes in the dark at normal temperatures. The annexin-V binding buffer was then added. The analysis of the cells was carried out utilizing FACSDiva software (BD Biosciences, San Diego, CA, USA). Data were shown as percentages of annexin-V^+^ lymphocytes. The positive threshold was constructed using non-stained controls.

### 2.6. Statistical Analyses

The data was analyzed using IBM SPSS statistical package software version 28 (IBM Corp., computer software, Armonk, NY, USA). For quantitative non-parametric data, descriptive statistics were computed using the interquartile range (IQR), median, and frequency, while percentages and numbers were employed for qualitative data. To compare quantitative non-parametric data from two different groups, the Mann–Whitney test was utilized, while Fisher’s exact or chi-squared analysis was performed to compare the two groups’ qualitative data. Pearson’s correlation was determined between the continuous variables. Using the receiver operator characteristic (ROC) curves, the proper cutoff point, specificity, sensitivity, NPV, PPV, and accuracy of the numerous variables were determined. The area under the curve (AUC) was utilized to investigate the prediction accuracy of NHL markers. The *p*-value cutoff was less than 0.05.

## 3. Results

### 3.1. Demographic and Clinical Criteria

The demographic and clinical criteria of the healthy controls and newly diagnosed NHL subjects are outlined in Table 1. The subjects were divided into twenty NHL patients and twenty normal volunteers. The NHL patients comprised ten males and ten females, with a mean age of 39.25 ± 16.2 years; their ages ranged from 27.5 to 49.5 years old. The twenty normal controls comprised ten males and ten females, with a mean age of 40.3 ± 15.9 years; their ages ranged from 25.5 to 50.0 years old. No significant difference was found between NHL subjects and normal volunteers regarding gender and age *(p* = 1.0 and *p* = 0.860, respectively).

Based on the clinical criteria of the NHL patients, 7 (35%) had hepatomegaly, and 9 (45%) had splenomegaly. There was a significant difference between hepatomegaly and splenomegaly when patients with NHL were compared with the normal controls (*p* = 0.008 and *p* = 0.001). In terms of lymph node enlargement, there was a significant difference between NHL patients and healthy volunteers (*p* < 0.001) (Table 1).

### 3.2. Circulating Lymphocytes in NHL

To assess the immune response to lymphomas, circulating lymphocytes in the NHL patients and normal controls were evaluated (Table 2). Compared with the healthy volunteers, the NHL patients demonstrated decreased percentages of CD3^+^ and CD3^+^CD4^+^ cells (median = 57.0 vs. 71.5 and 39.5 vs. 44.9, respectively; *p* < 0.001) and decreased percentages of CD3^+^CD8^+^ cells (median = 17.0 vs. 22.0; *p* = 0.027). No significant variations in CD20^+^ cells were seen between the normal volunteers and NHL patients (Table 2).

### 3.3. PD-L1^+^/PD-1 Upregulation in Circulating Lymphocytes in NHL Patients

The PD-L1 expression in circulating lymphocytes is illustrated in Table 3. Compared with the healthy volunteers, the NHL patients had significantly greater PD-L1^+^CD3^+^ and PD-L1^+^CD3^+^CD4^+^ percentages (median = 53.0 vs. 2.7 and 20.5 vs. 0.95, respectively; *p* < 0.001). PD-L1^+^CD3^+^CD8^+^ % and PD-L1^+^CD20^+^ % were statistically higher in the NHL subjects than in the controls (median = 8.1 vs. 0.5 and 16 vs. 1, respectively; *p* < 0.001).

The PD-1 expression in circulating lymphocytes was evaluated (Table 3). The NHL patients had higher PD-1^+^CD3^+^ and PD-1^+^CD3^+^CD4^+^ percentages compared with the normal volunteers (median = 9.5 vs. 2.0 and 7.0 vs. 0.8, respectively; *p* < 0.001). Moreover, there was a substantial difference in percentages of PD-1^+^CD20^+^ and PD-1^+^CD3^+^CD8^+^ cells between NHL patients and normal controls (median = 8.0 vs. 0.7 and 5.0 vs. 0.45, respectively; *p* < 0.001).

### 3.4. Expression of PD-L1/PD-1 in Apoptotic Lymphocytes

Lymphocyte apoptosis was investigated (annexin-V binding) by flow cytometry. NHL subjects had a higher percentage of annexin-V^+^PD-L1^+^ lymphocytes than the healthy controls (median = 40 vs. 0.5; *p* < 0.001). The annexin-V^+^PD-1^+^ percentage was statistically higher in the NHL subjects than in the normal volunteers (9.5 vs. 0.1; *p* < 0.001) (Figure 1).

Higher percentages of apoptotic CD4^+^ and CD8^+^ were observed in the NHL patients compared with the normal controls (median = 27.5% vs. 0.1% and 10.5% vs. 0.1%, respectively; *p* < 0.01). The CD20^+^ subpopulations with annexin-V were increased in the NHL subjects compared with the normal volunteers (median = 17.5% vs. 0.1%; *p* < 0.01) (Table 4).

### 3.5. Study of Peripheral Lymphocytes and PD-L1 Expression

The relationship between the markers in NHL patients was evaluated (Table 5). Annexin-V^+^CD3^+^ cells were positively associated with PD-1^+^CD3^+^ and PD-L1^+^CD3^+^ cells. Annexin-V^+^CD3^+^ lymphocytes had a positive association with PD-L1^+^CD8^+^ and PD-1^+^CD4^+^ lymphocytes (r = 0.615, *p* = 0.004 and r = 0.565, *p* = 0.009, respectively). Annexin-V^+^CD8^+^ lymphocytes had a positive association with PD-L1^+^CD8^+^ cells (r = 0.618, *p* = 0.004). Annexin-V^+^CD4^+^ lymphocytes had a significant correlation with PD-1^+^CD4^+^ lymphocytes (r = 0.477, *p* = 0.034). Annexin-V^+^CD20^+^ cells were positively associated with CD20^+^ and PD-L1^+^CD20^+^ cells (r = 0.968, *p* < 0.001 and r = 0.887, *p* < 0.001) (Table 5).

Annexin-V^+^ lymphocytes had a significant correlation with PD-1^+^CD3^+^ and PD-L1^+^CD3^+^ lymphocytes (r = 0.805, *p* < 0.001 and r = 0.593, *p* = 0.006, respectively). Moreover, annexin-V^+^ lymphocytes had a significant association with PD-L1^+^CD8^+^ and PD-1^+^CD4^+^ lymphocytes (r = 0.514, *p* = 0.020 and r = 0.598, *p* = 0.005). Annexin-V^+^PD-1^+^ lymphocytes positively correlated with annexin-V^+^PD-L1^+^ cells (*r* = 0.593, *p* = 0.006) (Appendix A).

The relationship between the markers and the clinical criteria of NHL lymphoma patients was evaluated (Appendix A). Annexin-V^+^PD-L1^+^ and annexin-V^+^PD-1^+^ lymphocytes were positively associated with hepatomegaly, but this did not match the significance (*p* > 0.05). Furthermore, High PD-L1/PD-1 expression was positively associated with hepatomegaly but without significance. Additionally, positive correlations were revealed between PD-L1^+^CD4^+^ and PD-L1^+^CD20^+^ lymphocytes and splenomegaly (*p* = 0.093 and *p* = 0.015) (Appendix A).

In NHL subjects, PD-L1^+^CD3^+^ lymphocytes were associated with PD-1^+^CD3^+^ cells (*p* = 0.008). PD-1^+^CD3^+^ lymphocytes correlated with PD-1^+^CD4^+^ lymphocytes, while PD-1^+^CD8^+^ lymphocytes correlated with PD-1^+^CD20^+^ lymphocytes (*p* = 0.013 and *p* = 0.003, respectively) (Appendix A).

### 3.6. Diagnostic Performance of Circulating Cells and PD-L1 Expression for the Identification of NHL Patents

A ROC analysis was performed for the immune cells to define the cutoff values that would balance the false-negative and false-positive rates with the best predictive level (Figure 2). The AUCs of the CD3^+^ and CD4^+^ percentages were 0.875 and 0.704 (*p* = 0.022 and *p* < 0.001), respectively, while the AUCs of CD8^+^ and CD20^+^ percentages were 0.808 and 0.588, respectively (*p* = 0.346). The ROC curves illustrated that a CD3^+^ percentage ≤ 63, a CD4^+^ percentage ≤ 38, a CD8^+^ percentage ≤ 18, and a CD20^+^ percentage ≤ 21 were the optimal cutoffs to identify NHL patients. The CD3^+^ percentage had 95% specificity values, and the sensitivity was 65%. The CD4^+^ and CD8^+^ percentages were 100% specific for the detection of NHL; the sensitivities were 45% and 55%, respectively. The sensitivity and specificity of the CD20^+^ percentage were 45% and 75%, respectively (*p* = 0.346) (Figure 2; Table 6).

The ROC analysis for the identification of NHL patients is illustrated in Table 6. The PD-L1^+^CD8^+^ and PD-L1^+^CD4^+^ AUCs were both 1 (*p* < 0.001). PD-L1^+^CD20^+^ had an AUC of 1 (*p* < 0.001). The cutoff levels for PD-L1^+^CD4^+^, PD-L1^+^CD8^+^, and PD-L1^+^CD20^+^ for the differentiation of NHL patients were >1.6, >1.1, and >1.1, respectively. The expression of PD-L1 in peripheral lymphocytes had 100% sensitivity and specificity values (*p* < 0.001).

## 4. Discussion

Lymphomas have different immune-evasion mechanisms, implying that the evasion of anti-tumor immunity is necessary for their pathogenesis [23]. The PD-1/PD-L1 signaling pathway inhibits lymphocyte activation and has a significant role in cancer immune evasion [24]. PD-L1 and PD-1 expression in tumors and tumor-infiltrating immune cells has previously been reported [10,25]. However, the PD-L1/PD-1 expression on circulating lymphocytes and its relationship to immunosuppression is unknown. This study assessed the expression of PD-L1 and PD-1 in the peripheral lymphocytes of NHL patients.

The results indicated that NHL patients had immune-function deficiencies in their peripheral blood, suggesting an immune defect. In agreement with previous reports, the study revealed that NHL patients had significantly decreased peripheral immune-cell percentages compared with the healthy controls [26,27,28,29,30,31,32,33]. When the immune system cannot function optimally, T and B cells fail to detect or destroy tumor cells [18,34]. A significant reduction in peripheral immune cells might be an essential parameter in the early detection of NHL.

The findings revealed that immune evasion could be mediated by mechanisms other than tumoral PD-L1 expression. The NHL patients had a significantly greater PD-L1 upregulation in circulating lymphocytes than the healthy controls; this was consistent with a previous study [32]. The mechanisms behind the PD-L1 increase in circulating immune cells are unknown. In lymphomas, chromosomal changes, genetic anomalies [35], and Epstein–Barr virus (EBV) infections can increase PD-L1 expression [23]. Another explanation is the inefficient recruitment of PD-L1^+^ cells into the tumor region, which increases the cell population in the peripheral blood. These results imply that PD-L1 might have a role in NHL pathogenesis, suggesting a diagnostic role of PD-L1 in NHL management and that a lymphoma-dependent existing state of immune suppression may be observed in these patients. Another scenario is that PD-L1 might reduce anti-tumor cytolysis in patients with NHL.

According to the study results, the PD-1 expression was upregulated in circulating lymphocytes but was negligible in normal volunteers; thus, PD-1 might have a role in NHL pathogenesis. The results demonstrated that the peripheral immune cells of patients had a low PD-1 expression. Fewer circulating PD-1^+^ cells may be a result of PD-1 loss in exhausted peripheral lymphocytes and fewer malignant cells expressing PD-1. A previous study reported a low expression of PD-1 in circulating CD20^+^ lymphocytes in DLBCL [32]. Zhang et al. [24] revealed a high PD-1 expression in CD4^+^ lymphocytes in DLBCL. These findings reveal that the relationship between CD4^+^, CD8^+^, and CD20^+^ cells and the PD-1 expression could identify the importance of immune cells in anti-tumor immunity.

Lymphocyte apoptosis is crucial for NHL etiology and pathogenesis [25]. T-lymphocyte functions are influenced by the PD-1/PD-L1 pathway via activation-induced cell death [10]. According to the results, the PD-L1 expression in circulating lymphocytes was dramatically upregulated in the NHL patients exhibiting lymphocyte apoptosis compared with the healthy volunteers. The percentages of annexin-V^+^PD-L1^+^ cells were three-fold higher than the annexin-V^+^PD-1^+^ percentages. These findings suggest that PD-L1 is an essential mediator of lymphocyte apoptosis, enhancing immunosuppression in NHL. Higher percentages of annexin-V^+^PD-L1^+^ cells suggest that PD-L1 might be the critical player in lymphoma pathogenesis. The study identified PD-L1 as a marker of NHL development by inhibiting peripheral immunity. These findings may be the outcome of the interaction between NHL and the immune system, confirming that enhanced PD-L1 upregulation as well as the downregulation of CD4^+^, CD8^+^, and CD20^+^ lymphocytes, can be detected in NHL. PD-L1 might aid NHL development by suppressing peripheral immune-cell activity. A more extensive study is required to confirm these results.

The study findings revealed a substantial link between annexin-positive lymphocytes and PD-L1/PD-1 expression. A higher annexin-V^+^ lymphocyte percentage had a significant association with increased PD-L1/PD-1^+^ lymphocytes. A strong association between lymphocyte apoptosis and PD-L1 was identified by annexin-V staining in the peripheral blood. The positive associations between lymphocyte apoptosis and the high PD-L1 expression in CD3^+^, CD8^+^, and CD20^+^ lymphocytes suggest that blocking PD-L1 would benefit lymphocyte production in NHL.

PD-1^+^CD3^+^ and PD-L1^+^CD3^+^ lymphocytes had a strong, significant positive relationship. A potential interaction between these molecules might explain this, as might the link between the peripheral cells and PD-L1/PD-1 in NHL. Another possibility is that PD-L1 is not cell-specific in NHL.

The best PD-L1 cutoff value is unknown and is antibody- and tumor-dependent [36,37,38]. As a result, discovering the appropriate PD-L1 cutoff value remains a challenge. For the first time, the baseline percentages of peripheral cells and PD-L1-expressing lymphocytes were used to differentiate patients at an increased risk of NHL. The findings suggested that the percentages of circulating lymphocytes and PD-L1^+^ lymphocytes could aid NHL prediction. The study persuaded that PD-L1 had high sensitivity and specificity for NHL prediction and, hence, might be a promising biomarker for the prediction and diagnosis of NHL. The study provides important information for clinicians to identify NHL patients at risk of immune suppression. This validates the utility of PD-L1 as a diagnostic tool for the detection of NHL patients at a higher risk of immune suppression. Future research, including a full clinical evaluation, might improve the understanding of PD-L1’s role in NHL.

The PD-1/PD-L1 inhibitors offer a significant application potential and therapeutic utility for the treatment of hematological cancers [17]. One of the most important consequences of fully understanding the effect of PD-1/PD-L1 on circulating lymphocytes is the application of novel therapies. PD-L1 inhibition can boost circulating lymphocyte functions, resulting in more effective disease eradication. Understanding the complexities of the involvement of PD-L1-targeted immunotherapy in circulating lymphocyte homeostasis is critical for the advancement of new therapies that protect immune cells and induce a strong anti-tumor response. Given the multidimensional nature of tumor–immunity interactions, prediction models relying on PD-L1 expression in circulating lymphocytes might be more applicable in the future.

The study is significant because it offers a new immunotherapeutic approach for various types of NHL, presenting a potential biomarker for predicting a diagnosis of NHL in patients. There were a few limitations in the study. First, the study’s sample size was small. A lack of PD-L1/PD-1 expression in cancer cells could be a study limitation. Future research examining tumor tissues and peripheral blood-matched specimens is required. To emphasize the significance of PD-L1/PD-1 in NHL etiology, future expression-level investigations are required on a broader geographical scale and with a greater number of patients.

## 5. Conclusions

This study revealed PD-L1/PD-1 upregulation in circulating lymphocytes in NHL. The quantitative fold difference between PD-L1 and PD-1 expression was statistically significant, implying that PD-L1 has diagnostic potential for NHL. PD-L1 upregulation in the circulating lymphocytes was associated with lymphocyte apoptosis. The study also offers a theoretical framework for using PD-L1 as a promising biomarker for NHL and immune suppression monitoring. To emphasize the significance of PD-L1 in NHL pathogenesis, research with a greater number of patients, multicenter collaborations, additional investigations, and clinical trials are required.

## Figures and Tables

**Figure 1 vaccines-11-01474-f001:**
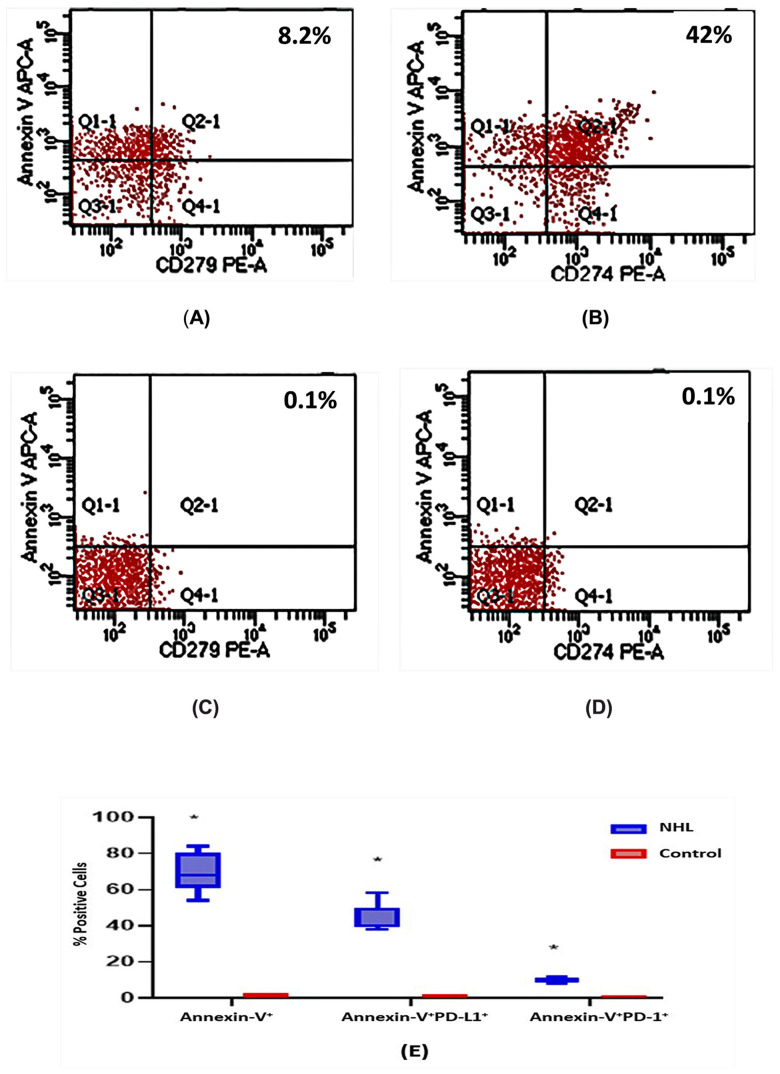
Lymphocyte apoptosis and expressions of PD-L1/PD-1 in apoptotic lymphocytes were investigated via flow cytometry: (**A**,**B**) upper-right quadrant represents gated annexin-V^+^PD-L1^+^ and annexin-V^+^PD-1^+^ lymphocytes in NHL subjects; (**C**,**D**) upper-right quadrant represents gated annexin-V^+^PD-L1^+^ and annexin-V^+^PD-1^+^ lymphocytes in normal controls; (**E**) diagram showing the significant difference between annexin-V, annexin-V^+^PD-L1^+^, and annexin-V^+^PD-1^+^ percentages of cells in NHL patients and normal controls. * *p* < 0.001.

**Figure 2 vaccines-11-01474-f002:**
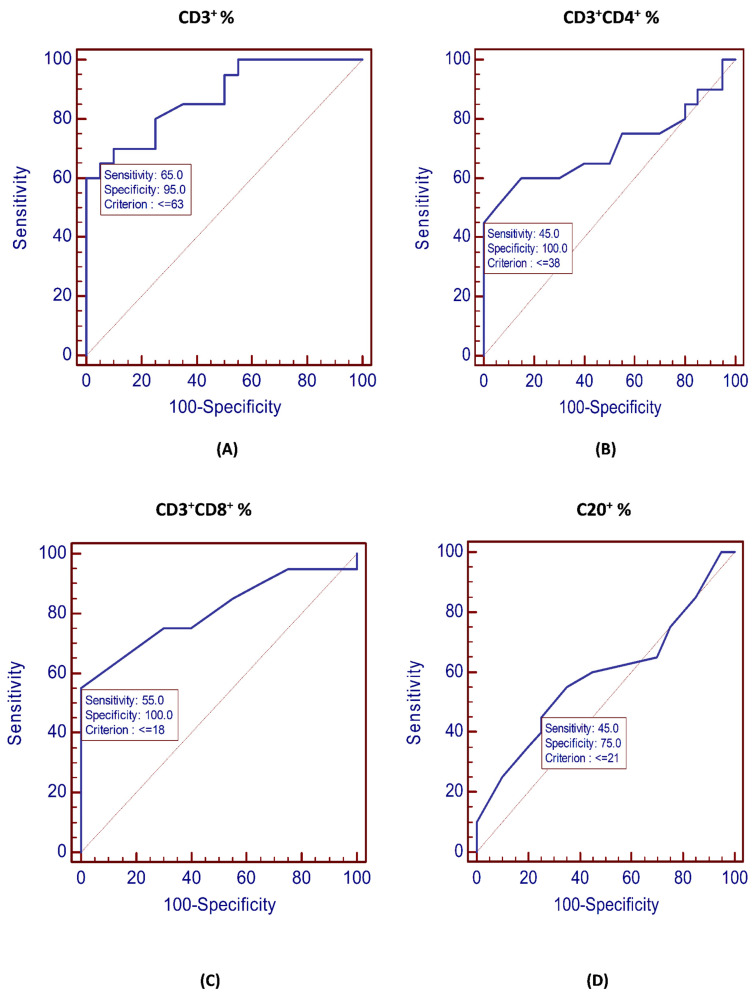
ROC curves for the discrimination of NHL patients: (**A**) CD3^+^ percentage; (**B**) CD3^+^CD4^+^ percentage; (**C**) CD3^+^CD8^+^ percentage; (**D**) CD20^+^ percentage.

**Table 1 vaccines-11-01474-t001:** Clinical and demographic characteristics of NHL patients and normal controls. NHL: non-Hodgkin lymphoma; mean ± SD: mean ± standard deviation; N: number. ** *p* < 0.001; * *p* < 0.05.

	NHL	Normal Controls	*p*-Value
	(N = 20)	(N = 20)	
Age (mean ± SD)	39.25 ± 16.2	40.3 ± 15.9	0.86
Sex			1.00
Male	10 (50.0%)	10 (50.0%)	
Female	10 (50.0%)	10 (50.0%)	
Hepatomegaly	7 (35.0%)	0 (0.0%)	0.008 *
Splenomegaly	9 (45.0%)	0 (0.0%)	0.001 **
Enlarged lymph node	20 (100.0%)	0 (0.0%)	<0.001 **

**Table 2 vaccines-11-01474-t002:** Median percentages of circulating lymphocytes in normal controls and NHL patients. IQR: interquartile range; NHL: non-Hodgkin lymphoma; N: number. ** *p* < 0.001, * *p* < 0.05.

		NHL	Normal Controls	Z_MWU_	*p*-Value
		(N = 20)	(N = 20)		
CD3^+^, %	Median	57 (42–68.2)	71.5 (67.5–77)	−4.061	<0.001 **
	IQR				
CD3^+^CD4^+^, %	Median	39.5 (32.5–46.25)	44.9 (43–47.5)	−2.207	0.027 *
	IQR				
CD3^+^CD8^+^, %	Median	17 (14–21)	22 (20–24.5)	−3.36	0.001 *
	IQR				
CD20^+^, %	Median	22 (18.5–25.5)	24 (21–25.5)	−0.952	0.341
	IQR				

**Table 3 vaccines-11-01474-t003:** PD-L1/PD-1 percentages in circulating lymphocytes in NHL patients and healthy controls. NHL: non-Hodgkin lymphoma; IQR: interquartile range. ** *p* < 0.001.

		NHL	Controls	*p*-Value
		(N = 20)	(N = 20)	
PD-L1^+^CD3^+^, %	IQR	53 (33.8–58.85)	2.7 (1.5–3.6)	<0.001 **
	Median			
PD-L1^+^CD3^+^CD4^+^, %	IQR	20.5 (16.5–22.5)	0.95 (0.4–1.25)	<0.001 **
	Median			
PD-L1^+^ CD3^+^CD8^+^, %	IQR	8.1 (6–11.2)	0.5 (0.4–0.8)	<0.001 **
	Median			
PD-L1^+^CD20^+^, %	IQR	16 (14–19)	1 (0.8–1)	<0.001 **
	Median			
PD-1^+^CD3^+^, %	IQR	9.5 (8–11.5)	2 (1.2–2.5)	<0.001 **
	Median			
PD-1^+^CD3^+^CD4^+^, %	IQR	7 (6–7.5)	0.8 (0.55–1.1)	<0.001 **
	Median			
PD-1^+^ CD3^+^CD8^+^, %	IQR	5 (4–6)	0.45 (0.35–0.8)	<0.001 **
	Median			
PD-1^+^CD20^+^, %	IQR	8 (7–9)	0.7 (0.6–0.9)	<0.001 **
	Median			

**Table 4 vaccines-11-01474-t004:** Lymphocyte apoptosis in NHL patients. Annexin-V binding in lymphocyte subsets in healthy individuals and NHL subjects. IQR: interquartile range; NHL: non-Hodgkin lymphoma; N: number. ** *p* < 0.001.

		NHL	Controls	*p*-Value
		(N = 20)	(N = 20)	
Annexin-V^+^CD3^+^, %	IQR	35 (25.5–45.5)	0.1 (0.05–0.15)	<0.001 **
	Median			
Annexin-V^+^CD3^+^CD4^+^, %	IQR	27.5 (23.5–39)	0.1 (0–0.1)	<0.001 **
	Median			
Annexin-V^+^CD3^+^CD8^+^, %	IQR	10.5 (9–12.5)	0.1 (0–0.1)	<0.001 **
	Median			
Annexin-V^+^CD20^+^, %	IQR	17 (16–22)	0.1 (0–0.1)	<0.001 **
	Median			

**Table 5 vaccines-11-01474-t005:** Correlation between percentages of annexin-V^+^CD3^+^CD4^+^, annexin-V^+^CD3^+^, annexin-V^+^CD3^+^CD8^+^, and annexin-V^+^CD20^+^ cells and the significant variables in NHL patients. * *p* < 0.05; ** *p* < 0.001.

	Annexin-V^+^CD3^+^, %		Annexin-V^+^CD4^+^, %		Annexin-V^+^CD8^+^, %		Annexin-V^+^CD20^+^, %	
	r	*p*-Value	r	*p*-Value	r	*p*-Value	r	*p*-Value
Annexin-V^+^CD3^+^CD4^+^, %	0.386	0.093						
Annexin-V^+^CD3^+^CD8^+^, %	0.69	0.001 *	0.088	0.713				
Annexin-V^+^CD20^+^, %	−0.336	0.147	0.132	0.579	−0.33	0.156		
Annexin-V^+^PD-L1^+^, %	0.678	0.001 *	0.313	0.179	0.646	0.002 *	−0.172	0.468
Annexin-V^+^PD-1^+^, %	0.707	<0.001 **	0.367	0.112	0.53	0.016 *	−0.32	0.169
PD-L1^+^CD3^+^, %	0.567	0.009 *	0.149	0.531	0.317	0.174	−0.167	0.482
PD-L1^+^CD3^+^CD4^+^, %	−0.191	0.419	0.117	0.625	−0.235	0.318	0.28	0.233
PD-L1^+^CD3^+^CD8^+^, %	0.615	0.004 *	0.084	0.724	0.618	0.004 *	−0.205	0.386
PD-L1^+^CD20^+^, %	−0.345	0.136	0.167	0.483	−0.379	0.099	0.887	<0.001 **
PD-1^+^CD3^+^, %	0.744	<0.001 **	0.343	0.138	0.588	0.006 *	−0.388	0.091
PD-1^+^CD3^+^CD4^+^, %	0.565	0.009 *	0.477	0.034 *	0.602	0.005 *	0.029	0.903
PD-1^+^CD3^+^CD8^+^, %	−0.019	0.938	0.413	0.07	0.344	0.137	0.09	0.705
PD-1^+^CD20^+^, %	−0.237	0.315	0.365	0.114	−0.082	0.73	−0.003	0.992
CD3^+^, %	0.945	<0.001 **	0.373	0.105	0.655	0.002 *	−0.317	0.174
CD3^+^CD4^+^, %	0.66	0.002 *	0.179	0.449	0.589	0.006 *	−0.218	0.356
CD3^+^CD8^+^, %	0.607	0.005 *	0.085	0.721	0.936	<0.001 **	−0.264	0.26
CD20^+^, %	−0.425	0.062	0.102	0.669	−0.422	0.064	0.968	<0.001 **

**Table 6 vaccines-11-01474-t006:** Diagnostic performances of peripheral lymphocytes and the PD-L1 expression in lymphocyte subsets in NHL patients. * *p* < 0.05; ** *p* < 0.001.

	Cutoff	AUC	*p*-Value	Sensitivity	Specificity
CD3^+^, %	≤63	0.875	<0.001 **	65%	95%
CD3^+^CD4^+^, %	≤38	0.704	0.022 *	45%	100%
CD3^+^CD8^+^, %	≤18	0.808	<0.001 **	55%	100%
CD20^+^, %	≤21	0.588	0.346	45%	75%
PD-L1^+^CD3^+^CD4^+^, %	>1.6	1.0	<0.001 **	100%	100%
PD-L1^+^CD3^+^CD8^+^, %	>1.6	1.0	<0.001 **	100%	100%
PD-L1^+^CD20^+^, %	>1.1	1.0	<0.001 **	100%	100%

## Data Availability

The data analyzed during this work can be requested from the author.

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
