# Peer review of "PD-L1 Is Involved in the Development of Non-Hodgkin’s Lymphoma by Mediating Circulating Lymphocyte Apoptosis"

_vaccines, 2023, doi:10.3390/vaccines11091474_

Round 1

Reviewer 1 Report

1. In the introduction, lines 28-29, the author states that Recently, significant insights into the interactions between the immune system and tumor cells have been attained. However, he does not describe what the significant insights are. The author must describe these significant insights. Reference 4 is very specific and is related only to treatment. The author must change the reference to a more appropriate one(s) that support her assertion.

2. Lines 35-36 refer to the existence of previous reports of PD-1/PD-L1 in the NHL. However, just cite the work in reference 12, which not only addresses NHL, but also another HL, among others. Authors are recommended to review the work of Gravelle et al (PMID: 28402953 PMCID: PMC5546533 DOI: 10.18632/oncotarget.16680), which describes a summary of studies of PD-1/PD-L1 expression in non-Hodgkin lymphoma.

3. In the discussion, lines 232-234, the author states that there are previous reports revealing that NHL patients, compared to healthy controls, have a decreased percentage of immune cells. However, he only cites the work in reference 24. The author must cite other works that support his statement, or else refer to the only previous report 24.

4. In the discussion, lines 239-242, the author states that his results are consistent with a few other studies. However, he only cites reference 27. The author must cite the other studies consistent with his results, or simply state that his results are consistent with those obtained in reference 27.

Author Response

Reviewer 1

Editing of the English language, style, and layout was performed by a MDPI editor.

All sections of MS are improved.

Comment 1- In the introduction, lines 28-29, the author states that Recent insights into the interactions between the immune system and tumor cells have been attained. However, he does not describe what the significant insights are. The author must describe these significant insights. Reference 4 is very specific and is related only to treatment. The author must change the reference to a more appropriate one(s) that supports her assertion.

Response:  Thanks much for this valuable comment. The significant insights into the interactions between the immune system and tumor cells are described with the change of reference 4.

Significant insights into the interactions between the immune system and lymphomas have been attained [3]. Certain infections and immune deficiencies may play a role in NHL pathogenesis [4]. On the other hand, Lymphomas cause immunological alterations by lowering B lymphocytes and interrupting the regular interaction of B lymphocytes with other cells [5]. Immune-system silencing initiates the development and growth of NHL [6]. Immunosuppressive networks are characterized by cellular microenvironmental alterations and different signaling pathways [7]. The expression of programmed death-ligand 1 (PD-L1)/PD-1 could enhance tumor development via the inhibition of T-cell functions, indicating an immunosuppressive role of PD-L1/PD-1 [8–10].

Comment 2- Lines 35-36 refer to previous reports of PD-1/PD-L1 in the NHL. However, citing the work in reference 12, which addresses not only NHL but also another HL, among others. Authors are recommended to review the work of Gravelle et al. (PMID: 28402953 PMCID: PMC5546533 DOI: 10.18632/oncotarget.16680), which describes a summary of studies of PD-1/PD-L1 expression in non-Hodgkin lymphoma.

Response 2: Thanks for this valuable comment. The work cited by Gravelle et al. in reference 15.

Comment 3- In discussion lines 232-234, the author states that previous reports reveal that NHL patients, compared to healthy controls, have a decreased percentage of immune cells. However, he only cites the work in reference 24. The author must cite other works that support his statement or else refer to the only previous report 24.

Response 3:  Thanks for this valuable comment; other works are cited to support the statement. In agreement with previous reports, the study revealed that NHL patients had significantly decreased peripheral immune-cell percentages compared with the healthy controls [26–33].

Comment 4- In discussion lines 239-242, the author states that his results are consistent with a few other studies. However, he only cites reference 27. The author must cite the other studies consistent with his results or state that his results are consistent with those obtained in reference 27.

Response 4:  Thanks much for this valuable comment; this is corrected.

The NHL patients had a significantly greater PD-L1 upregulation in peripheral lymphocytes than the healthy controls; this was consistent with a previous study [32].

Reviewer 2 Report

This article is on PD-L1`s involvement in the development of Non-Hodgkin’s Lymphoma. The topic could be of interest for readers with a specific interest in aspects related to hematological malignancies. However, the manuscript needs thorough editing to enhance its comprehensibility and to comply with standards for scientific writing and ethics.

I would suggest adding some more meaningful keywords to the list.

There is a copy paste hint for hematological malignancy which is a hyperlink to a definition. Thus, I suggest checking for plagiarism in the article.

The protocol as described in the methods section is superficial as the recruitment of these study participants and ethical considerations are unclear and the exact concentrations and producer of the components.

The participants mean age and SD should be presented.

The format of the tables is not homogenous.

Figure 1 is incomplete.

There are several typos and formatting issues such as in line 104 or “ROC curves for Discriminating patient” or “that Immune evasion”. These are only some examples, as there are many more errors in the manuscript.

The practical and theoretical implications of the study findings are not discussed in great detail, and the conclusion is very short and vague and could be more meaningful.

Author Response

Reviewer 2

Editing of the English language, style, and layout was performed by a MDPI editor.

All sections of MS are improved.

Comment 1-I would suggest adding some more meaningful keywords to the list.

Response 1:  Thanks for this valuable comment. some more meaningful keywords to the list are added. non-Hodgkin lymphoma; programmed death-ligand 1; apoptosis; circulating lymphocytes.

Comment 2- There is a copy-paste hint for hematological malignancy which is a hyperlink to a definition. Thus, I suggest checking for plagiarism in the article.

Response 2:  Thanks for this valuable comment. Hematological malignancy statement is reviewed. Non-Hodgkin’s lymphoma (NHL) tumors are the most prevalent hematological tumors in the world and constitute a heterogeneous entity [1]. The WHO has classified over eighty distinct NHL subtypes based on the cell type (NK, T, and B cells) and the clinical characteristics, genetic criteria, immunophenotyping, and morphology [2].

Comment 3- The protocol as described in the methods section is superficial as the recruitment of these study participants and ethical considerations are unclear and the exact concentrations and producer of the components.

Response 3:  Thanks much for this valuable comment. The subjects and methods section is greatly improved. the recruitment of the study participants and ethical considerations are provided. The exact concentrations and producers of the components are included.

Comment 4- The participant’s mean age and SD should be presented.

Response 4:  Thanks for the reviewer. The participant’s mean age and SD are presented in Table 1.

Comment 5- The format of the tables is not homogenous.

Response 5:  Thanks for this valuable comment. The format of the tables is homogenous.

Comment 6- Figure 1 is incomplete.

Response 6:  Thanks for this valuable comment. Figure 1e is complete.

Comment 7- There are several typos and formatting issues such as in line 104 “ROC curves for Discriminating patient” or “that Immune evasion”. These are only some examples, as there are many more errors in the manuscript.

Response 7:  Thanks much for this valuable comment. Typos and formatting issues errors are corrected. Errors are all corrected.

Comment 8- The practical and theoretical implications of the study findings are not discussed in great detail, and the conclusion is very short and vague and could be more meaningful.

Response 8:  - Thanks much for this valuable comment. The practical and theoretical implications of the study findings are discussed in detail and the conclusion is changed and more meaningful.

        The best PD-L1 cutoff value is unknown and is antibody- and tumor-dependent [39–41]. As a result, discovering the appropriate PD-L1 cutoff value remains a challenge. For the first time, the baseline percentages of peripheral cells and PD-L1-expressing lymphocytes were used to differentiate patients at an increased risk of NHL. The findings suggested that the percentages of peripheral lymphocytes and PD-L1+ lymphocytes could aid NHL prediction. The study persuaded that PD‑L1 had high sensitivity and specificity for NHL prediction and hence might be a promising biomarker for the prediction and diagnosis of NHL. Our study provides important information for clinicians to identify NHL patients at risk of immune suppression. This validates the utility of PD-L1 as a diagnostic tool for the detection of NHL patients at a higher risk of immune suppression. Future research, including a full clinical evaluation, might improve the understanding of PD-L1's role in NHL.

       The PD-1/PD-L1 inhibitors offer a significant application potential and therapeutic utility for the treatment of human tumors. One of the most important consequences of fully understanding the effect of PD-1/PD-L1 on peripheral lymphocytes is the application of novel therapies. PD-L1 inhibition can boost peripheral lymphocyte functions, resulting in more effective disease eradication. Understanding the complexities of the involvement of PD-L1-targeted immunotherapy in peripheral lymphocyte homeostasis is critical for the advancement of new therapies that protect immune cells and induce a strong anti-tumor response. Given the multidimensional nature of tumor–immunity interactions, prediction models relying on PD-L1 expression in peripheral lymphocytes might be more applicable in the future.

   5. Conclusions

      This study revealed PD-L1/PD-1 upregulation in peripheral lymphocytes in NHL. The quantitative fold difference between PD-L1 and PD-1 expression was statistically significant, implying that PD-L1 has diagnostic potential for NHL. PD-L1 upregulation in the circulating lymphocytes was associated with lymphocyte apoptosis. The data imply that PD-L1 could be beneficial as a biomarker for NHL and immune suppression monitoring. To emphasize the significance of PD-L1 in NHL pathogenesis, research with a greater number of patients, multicenter collaborations, additional investigations, and clinical trials are required. the study also offers a theoretical framework for the use of PD-L1 inhibitors in NHL, but this must be confirmed by large, well-designed cohort studies.

Round 2

Reviewer 2 Report

I thank the author for revising the manuscript. There a still some issues with the paper

For example, 2. Subjects and Methods should be renamed to 2. Methods.

Please indicate, in which country the university is located.

SPSS: citation is needed.

Table 3 seems to have 2 parts.

The conclusion stated that a theoretical framework was provided, which is however not presented in the results.

Is it correct that "This work was published in Vaccines with the patient's full written consent. "? This statement is very uncommon and seems to contradcit good scientific practice.

Author Response

Response to Reviewer 2

Thank you for allowing to submit a revised form of " PD-L1 is involved in the development of Non-Hodgkin’s Lymphoma by mediating Circulating Lymphocyte Apoptosis " for publication in the Journal of Vaccines. Thanks much for the insightful remarks and valuable modifications to the manuscript. The reviewer's suggestions were performed and highlighted in the paper.

Comment 1-Subjects and Methods should be renamed to 2. Methods.

Response 1:  Thanks for this valuable comment. Subjects and Methods were renamed to Methods.

Comment 2- Please indicate in which country the university is located.

Response 2:  Thanks for this valuable comment. The country (Egypt) in which the university is located was put in the methods section. The study was conducted at the Clinical Pathology Department of the Faculty of Medicine, Minia University, Egypt.

Comment 3- SPSS: citation is needed.

Response 3:  Thanks much for this valuable comment. SPSS: citation is added in the Statistical Analyses. The data was analyzed using IBM SPSS statistical package software version 28 (IBM Corp., computer software, Armonk, New York, USA)

Comment 4- Table 3 seems to have 2 parts.

Response 4:  Thanks for the reviewer. Table 3 is corrected to be one part.

Comment 5- The conclusion stated that a theoretical framework was provided, which is however not presented in the results.

Response 5:  Thank you for pointing this out. Previous literature assumed that PD-L1 overexpression, association of PD-L1 expression with poor prognostic criteria, and predictive models based on expression of PD-L1, might provide a theoretical foundation for the future application of PD-L1 inhibitors in the cancer era (Shen et al. World Journal of Surgical Oncology (2019) 17:4; Tang et al. Frontiers in Immunology (2022) Volume 13 https://doi.org/10.3389/fimmu.2022.964442. Although, I appreciate and agree with the reviewer's comment. Accordingly, throughout the conclusion, I have deleted the sentence and replaced it with (The study also offers a theoretical framework for using PD-L1 as a promising biomarker for NHL and immune suppression monitoring).

Comment 6- Is it correct that "This work was published in Vaccines with the patient's full written consent. "? This statement is very uncommon and seems to contradict good scientific practice.

Response 6: Thank you for this comment. It is interesting to explore this aspect. This statement is found in some situations and does not mean contradiction to good scientific practice. The manuscript (PD-L1 is involved in the development of Non-Hodgkin’s Lymphoma by mediating Circulating Lymphocyte Apoptosis) is a research study involving NHL patients and controls. Sometimes, data availability is requested, and hence the identifying information of the subjects. In my manuscript (Endothelial Monocyte-Activating Polypeptide-II Is an Indicator of Severity and Mortality in COVID-19 Patients. Vaccines (Basel). 2022 Dec; 10(12): 2177), the statement was involved (the patient’s informed written consent was obtained to publish this paper in Vaccines).